# Countercurrent heat exchange and thermoregulation during blood-feeding in kissing bugs

Chloé Lahondère[1§], Teresita C Insausti[1], Rafaela MM Paim[2†], Xiaojie Luan[3†], George Belev[4], Marcos H Pereira[2], Juan P Ianowski[3‡], Claudio R Lazzari[1‡*]

[1]Institut de Recherche sur la Biologie de l'Insecte, UMR 7261 CNRS - Université François Rabelais, Tours, France; [2]Instituto de Ciências Biológicas, Universidade Federal de Minas Gerais, Belo Horizonte, Brazil; [3]Department of Physiology, College of Medicine, University of Saskatchewan, Saskatoon, Canada; [4]Canadian Light Source Inc., Saskatoon, Canada

**Abstract** Blood-sucking insects experience thermal stress at each feeding event on endothermic vertebrates. We used thermography to examine how kissing-bugs *Rhodnius prolixus* actively protect themselves from overheating. During feeding, these bugs sequester and dissipate the excess heat in their heads while maintaining an abdominal temperature close to ambient. We employed a functional-morphological approach, combining histology, μCT and X-ray-synchrotron imaging to shed light on the way these insects manage the flow of heat across their bodies. The close alignment of the circulatory and ingestion systems, as well as other morphological characteristics, support the existence of a countercurrent heat exchanger in the head of *R. prolixus*, which decreases the temperature of the ingested blood before it reaches the abdomen. This kind of system has never been described before in the head of an insect. For the first time, we show that countercurrent heat exchange is associated to thermoregulation during blood-feeding.
DOI: https://doi.org/10.7554/eLife.26107.001

**\*For correspondence:**
claudio.lazzari@univ-tours.fr

[†]These authors contributed equally to this work
[‡]These authors also contributed equally to this work

**Present address:** [§]Department of Biology, University of Washington, Seattle, United States

**Competing interests:** The authors declare that no competing interests exist.

## Introduction

The entire life of insects highly depends on the environmental temperature ($T_a$) that impacts their body temperature ($T_b$), thus influencing their behaviour and physiology. Each species possesses a temperature range within which individuals can remain active; but within that range, their performance (e.g. food seeking or mating) is still temperature-sensitive (*Huey and Stevenson, 1979*). At temperatures below or above the activity range, insects can potentially experience thermal stress causing drastic physiological effects and even death. Nevertheless, different traits have evolved to avoid or minimise the effect of thermal stress due to extreme temperatures. In poikilotherms (i.e. $T_b = T_a$), these strategies include behavioural (e.g. postural control of exposition to solar radiation), biochemical [e.g. *heat-shock proteins* (HSPs) synthesis], and physiological (e.g. diapause) adjustments. However, some of them exhibit morpho-physiological adaptations that allow them regulating their own $T_b$ independently of $T_a$ (*Heinrich, 1993*; *May, 1979*).

If many insect species can avoid thermal stress simply by changing their posture or hide from sun exposure, some others, such as haematophagous (blood-feeding) insects, must expose themselves to high temperatures and their deleterious effects during blood-feeding (*Lahondère and Lazzari, 2012*). Indeed, most of them feed on endothermic vertebrates (e.g. birds) whose blood temperature can be as high as 40°C. On the other hand, in order to escape the defensive reactions of hosts against biting, they must ingest as much blood as possible, as fast as they can. Thus, they must cope with the rapid entry of a large volume of warm fluid, which temperature ($T_{blood}$) can greatly exceeds

**eLife digest** Many insect species have adopted the blood of birds and mammals as their main or even only food. Yet, blood is not freely available in nature, but it circulates inside vessels hidden under the skin of animals much bigger than the insect and capable of defending themselves from getting bitten. To succeed in getting a meal, blood-sucking insects must be able to feed quickly and take in as much blood as possible. Each time that they do this, a huge amount of warm fluid enters their body in just a few minutes. The blood temperature can be up to 20° or 25°C warmer than the insect itself. Moreover, an insect called a kissing bug may ingest up to 10 times its own weight in only fifteen minutes. The consequence is overheating and potentially harmful thermal stress.

Kissing bugs do not seem to suffer any harmful consequence of taking massive meals from warm-blooded animals. But why? The answer was unexpected: they simply do not warm up when they take a blood meal. However, it was not known how they manage to cool down the ingested blood.

By combining classical methods of studying anatomy with state of the art technologies, Lahondère et al. discovered that kissing bugs possess a sophisticated heat exchanger inside their heads. It works by transferring the heat associated with the ingested blood to the haemolymph (insect blood); these fluids circulate in opposite directions inside ducts that are close to each other in the head.

The discovery of a new system used by insects to cope with thermal stress expands our knowledge of insect physiology and opens new lines of research. The kissing bug heat exchanger could also serve as inspiration for equivalent technological systems. Last but not least, kissing bugs spread the parasites that cause Chagas disease in the Americas. Finding ways to disrupt the heat exchanger could prevent kissing bugs from feeding on blood, and so help to control the spread of disease.

DOI: https://doi.org/10.7554/eLife.26107.002

their own body temperature. This potentially generates thermal stress, and some species respond by synthesising HSPs, a cellular protective mechanism (*Benoit et al., 2011*; *Paim et al., 2016*). Other blood-feeding insects use thermoregulatory processes to protect themselves from overheating. For instance, *Anopheles* mosquitoes decrease their abdominal temperature by using evaporative cooling. They emit through the anus and then keep at the tip of their abdomen a droplet of fluid composed of urine and fresh blood, which evaporates cooling the abdomen, thus reducing the thermal shock and the associated stress caused by the blood-meal (*Lahondère and Lazzari, 2012*).

Haematophagous Diptera such as mosquitoes and tsetse flies can obtain a full blood-meal in less than 2 min (*Lehane, 2005*; *Lahondère and Lazzari, 2015*). The duration of their exposure to excessive heat is thus much shorter than kissing bugs, for example, that need between 15 and 20 min to feed to repletion. Moreover, kissing bugs can take blood meals that can attain up to 10 times their own unfed weight (*Lehane, 2005*). So, the heat gain at each feeding event is much higher in bugs than in mosquitoes or flies. Interestingly, it has been demonstrated that heating is deleterious for the physiology of kissing bugs. When they are exposed to a sub-lethal temperature of 30°C, bugs show delayed moulting, sterility and changes in their respiratory metabolism (*Okasha, 1968a*, *Okasha, 1968b*, *Okasha, 1968c*, *Okasha, 1968d 1970*; *Okasha et al., 1970*). Feeding on hot blood induces the synthesis of HSPs (*Paim et al., 2016*), but the increase in their expression level is relatively low compared to other insects (*Benoit et al., 2011*)

In the present work, we studied how the major Chagas disease vector *Rhodnius prolixus* copes with the excess of heat associated to blood-feeding. We performed real-time thermography and measured HSP expression, to analyse the extent to which these bugs are exposed to heat stress during feeding. Based on these results, we then performed a morpho-functional analysis of the *R. prolixus* head to gain insights on its structural organisation and how it could be implicated in heat management. Finally, we used synchrotron-based high-resolution X-ray imaging to see how the ingestion pumps work and to assess their role in blood displacement inside the insect. Based on the results obtained during this study, we propose a 3-D (three-dimensional) model of the *R. prolixus* head, as well as the existence of a countercurrent heat exchange mechanism through which this species can protect itself against heat stress during blood-feeding.

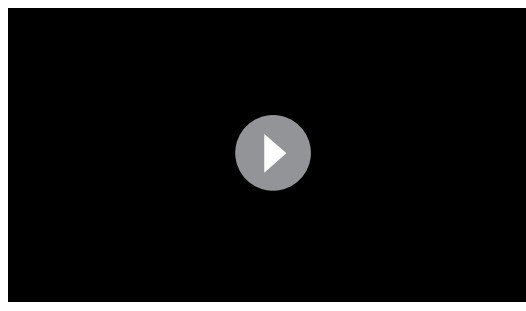

**Video 1.** *Rhodnius prolixus* thermogram during blood feeding on an artificial feeder. The blood was maintained at 37°C and the room temperature was 22°C. We recorded a frame every 10 s.
DOI: https://doi.org/10.7554/eLife.26107.003

## Results

### Real-time infrared thermography during feeding

To understand how *R. prolixus* manages the heat flow associated with the ingestion of a blood-meal, we first performed a real-time thermographic analysis of the dynamics of body warming during the entire feeding process (*Figure 1A*; SI *Video 1*). Before the blood intake, $T_b = T_a$ for all body parts of the insect. But once the insect started to feed, the different parts of its body did not exhibit the same temperature (i.e. heterothermy) (*Figure 2A–B*). Indeed, while the temperature of the proboscis ($T_p$) was high and close to the temperature of the blood, that of the abdomen ($T_{abd}$) was close to the ambient temperature, whereas the temperature of the head ($T_h$) and that of the thorax ($T_{th}$) remained intermediate. Thus, a marked thermal gradient was established along the insect body: $T_p > T_h > T_{th} > T_{abd}$. Interestingly, $T_p$ and $T_h$ oscillated during feeding, certainly in concordance with the variations in the activity of the ingestion pumps, while $T_{th}$ and $T_{ab}$ did not, remaining stable during the entire blood intake. These results show that *R. prolixus* is able to minimise the amount of heat reaching the abdomen during feeding (*Figure 2A–B*).

We then characterised regional heterothermy in *R. prolixus* by establishing different combinations of both ambient and blood temperature, to see how the temperature of the different parts of the body would be affected during blood-feeding (*Figure 1B*). At $T_a$ = 16°C, we observed a clear thermal gradient along the insect bodies either when fed on blood at 32°C or 37°C (*two-way* ANOVA, p-values<0.01 for all comparisons, see *Table 1*, *Figure 2C*, SI *Figure 2—figure supplement 1*, SI *Table 1*). At $T_{blood}$ = 37°C, the proboscis was 14°C warmer than the abdomen. A temperature gradient between the proboscis and the abdomen was also found in insects fed with blood at 32°C, 37°C and 42°C for all the different other $T_a$ conditions. Moreover, as $T_a$ increased, $T_b$ increased, but we noted that the temperature of the proboscis was more influenced by $T_{blood}$, while the abdomen

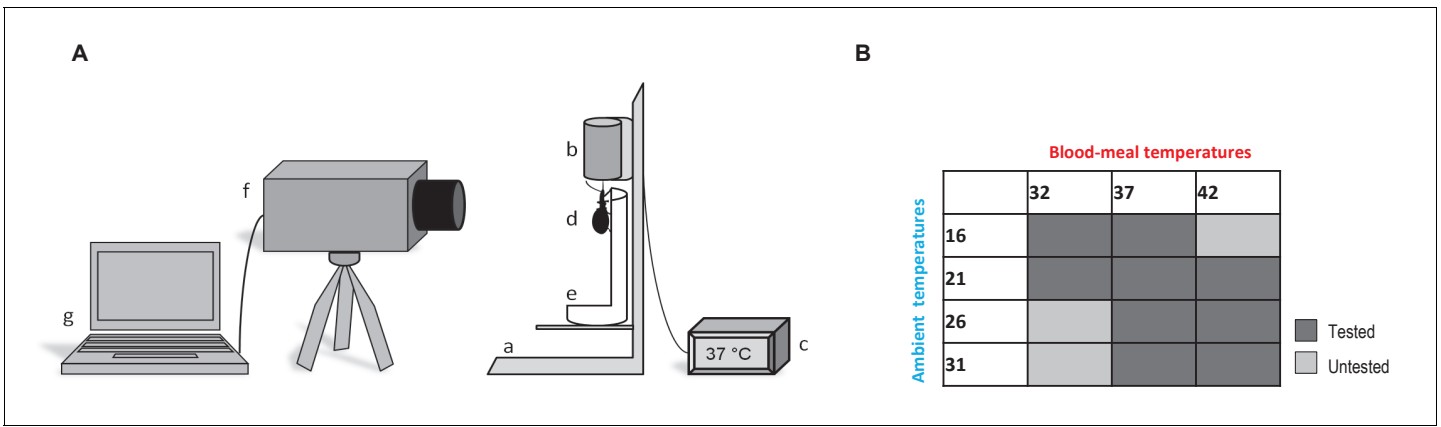

**Figure 1.** Thermographic experiments. (A) Experimental device used for *Rhodnius prolixus* blood-feeding and thermography. (a) artificial feeder, (b) blood container, (c) thermostat, (d) bug, (e) opened falcon tube, (f) thermographic camera, (g) computer. (B) Blood-meal and ambient temperatures combinations used during the experiments. Dark grey is related to the tested combinations; light grey corresponds to the combinations that were not tested.
DOI: https://doi.org/10.7554/eLife.26107.004

The following figure supplement is available for figure 1:

**Figure supplement 1.** General view (left panel) and details (right panel) of the setup used for infrared thermography.
DOI: https://doi.org/10.7554/eLife.26107.005

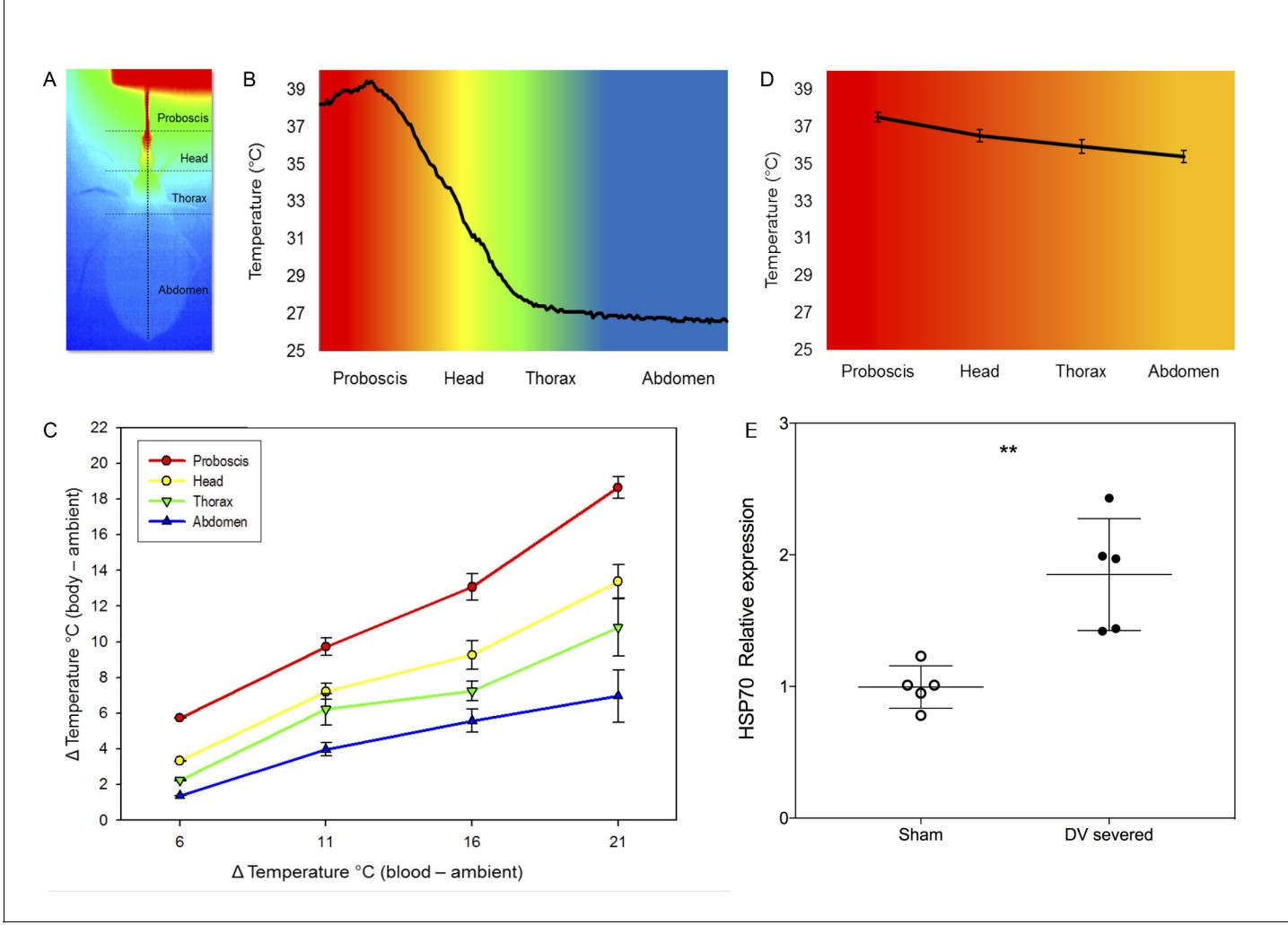

**Figure 2.** Heterothermy and HSP synthesis associated to feeding in kissing bugs. (**A**) Thermogram (dorsal view) of *Rhodnius prolixus* during feeding on sheep blood ($T_{blood}$ = 40°C) in an artificial feeder (on top). Horizontal dotted lines separate the different parts of the body, that is proboscis, head, thorax and abdomen. The vertical dotted line represents a transect along which the temperature of the surface of the insect's body was measured. (**B**) Temperatures recorded along the transect designated in A. ($T_{blood}$ = 40°C). (**C**) Impact of the difference between blood ($T_{blood}$) and environmental ($T_a$) temperatures on the body temperature of *Rhodnius prolixus*. (**D**) Average temperature of the proboscis, head, thorax and abdomen during blood-feeding in insects with severed dorsal vessels. (**E**) Relative levels of expression of HSP70 of sham-operated and dorsal vessel-severed bugs, measures 2 hr after fed on blood at 39°C. Asterisks: t-test, p<0.01, n = 5.

DOI: https://doi.org/10.7554/eLife.26107.006

The following source data and figure supplement are available for figure 2:

**Source data 1.** Source data of temperature in different parts of the body (P, proboscis; T, head; Th, thorax and A, abdomen) for different combinations of environment (16°C, 21°C, 26°C and 31°C) and blood temperature (32°C, 37°C and 42°C).

DOI: https://doi.org/10.7554/eLife.26107.008

**Source data 2.** Source data and statistical analysis (t-test) of the expression of HSP70 and HSP90 in bugs which dorsal vessel was severed before feeding and sham operated animals.

DOI: https://doi.org/10.7554/eLife.26107.009

**Figure supplement 1.** Mean temperatures of the proboscis (red), the head (yellow), the thorax (green) and the abdomen (blue) of *Rhodnius prolixus* during feeding on blood at either 32°C, 37°C or 42°C in an environment at either 16°C, 21°C, 26°C or 31 ± 1°C for all tested temperatures.

DOI: https://doi.org/10.7554/eLife.26107.007

tended to be more stable and did not vary as much, even if the temperature of ingested blood was higher (*Figure 2C*, *Figure 2—figure supplement 1*, *Figure 2—source data 1*). Our results showed that the different body regions warmed up differentially as a function of both the environment and blood temperatures (*two-way* ANOVA, p-values<0.01 for all comparisons, *Table 1*) and that the

**Table 1.** *Two-way* ANOVA table for the analysis of the impact of the temperature of the blood ($T_{blood}$) and the environmental temperature on the different body part of *Rhodnius prolixus* during feeding.
(See also *Figure 2C* and *Supplementary file 1*).

| Source of variation | DF | Sum of squares | Mean square | F-value | P-value |
|---|---|---|---|---|---|
| 1- Body part | 3 | 813.546 | 271.182 | 302.798 | <0.001 |
| 2- Tblood | 3 | 672.914 | 224.305 | 250.455 | <0.001 |
| 1 × 2 | 9 | 91.142 | 10.127 | 11.308 | <0.001 |
| Residual | 92 | 82.394 | 0.896 | | |
| Total | 107 | 1860.120 | 17.384 | | |

DOI: https://doi.org/10.7554/eLife.26107.010

temperature of the proboscis was close to $T_{blood}$, whereas the abdomen ($T_{abd}$) remained close to the temperature of the environment ($T_a$). In between the temperature gradient was not linear, but it exhibited a clear decrease at the posterior region of the head (*Figure 2B*), revealing that this region is particularly involved in heat transfer.

In order to examine the role of haemolymph circulation in heterothermy, we analysed the body temperature during blood-feeding of insects with severed dorsal vessels. As a result of the interruption of haemolymph circulation, heterothermy was greatly reduced. The thermal gradient observed was largely reduced in this group of insects ($T_p$ (37.5 ± 0.25°C), $T_h$ (36.5 ± 0.33°C), $T_{th}$ (35.92 ± 0.37°C) and $T_{abd}$ [35.38 ± 0.32°C]) in comparison to intact insects (*Figure 2D*). The mean temperature of the proboscis did not statistically differ from control bugs (Student *t*-test, n = 5, n.s), but the comparison of other regions, that is head tip, between eyes, thorax and centre of the abdomen, revealed significant differences (p-values<0.001 in all cases).

## HSP expression and thermal stress

To evaluate to what extent haemolymph circulation could be involved in reducing thermal stress during feeding, we compared the expression of HSP70 and HSP90 in insects with either intact or severed dorsal vessel. Whereas HSP90 did not evince a significant differential expression, HSP70 expression was significantly higher in insects with a severed dorsal vessel, a condition that impeded countercurrent (*Figure 2E*, *Figure 2—source data 2*, Student *t*-test p<0.01 two-tailed).

## Morphology and structural organisation of the head

Because heat seemed to keep confined in the head during blood-feeding, we then performed an analysis of functional morphology, in order to disentangle the underlying mechanism allowing *R. prolixus* to avoid abdominal warming during feeding. The head of *R. prolixus*, like other triatomine bugs, has a tubular shape, which shrinks a little anteriorly at the basis of the antennae and on its posterior part, just before the thorax. The brain is confined to the posterior region of the head, whereas the anterior region lodges the large cibarial pump musculature. Besides, there are large areas of haemocoel, both dorsally and ventrally to the nervous system, where haemolymph circulates (*Figure 3*).

### Circulatory system: aorta and haemolymphatic sinus

Behind the brain, the aorta (about 170 × 70 µm in section) becomes closely associated with the dorsal wall of the oesophagus (*Figure 3B–E*, *Figure 4C–H*). The aorta goes through the brain in the same configuration, almost surrounding the alimentary canal. On the anterior part of the head, forward from the optic lobes, the aorta reduces its diameter to about 50–75 µm, drops slightly and it turns into an haemolymphatic sinus (*Figure 3B–E*). This large sinus is delimited laterally by the muscles of the cibarial pump, ventrally by a thin membrane and dorsally by the head integument (*Figure 3B–E*, *Figure 4A–B*). Consequently, the haemolymph that comes from the abdomen, after crossing the brain, is projected in quite a large space separated from the outside only by the head wall.

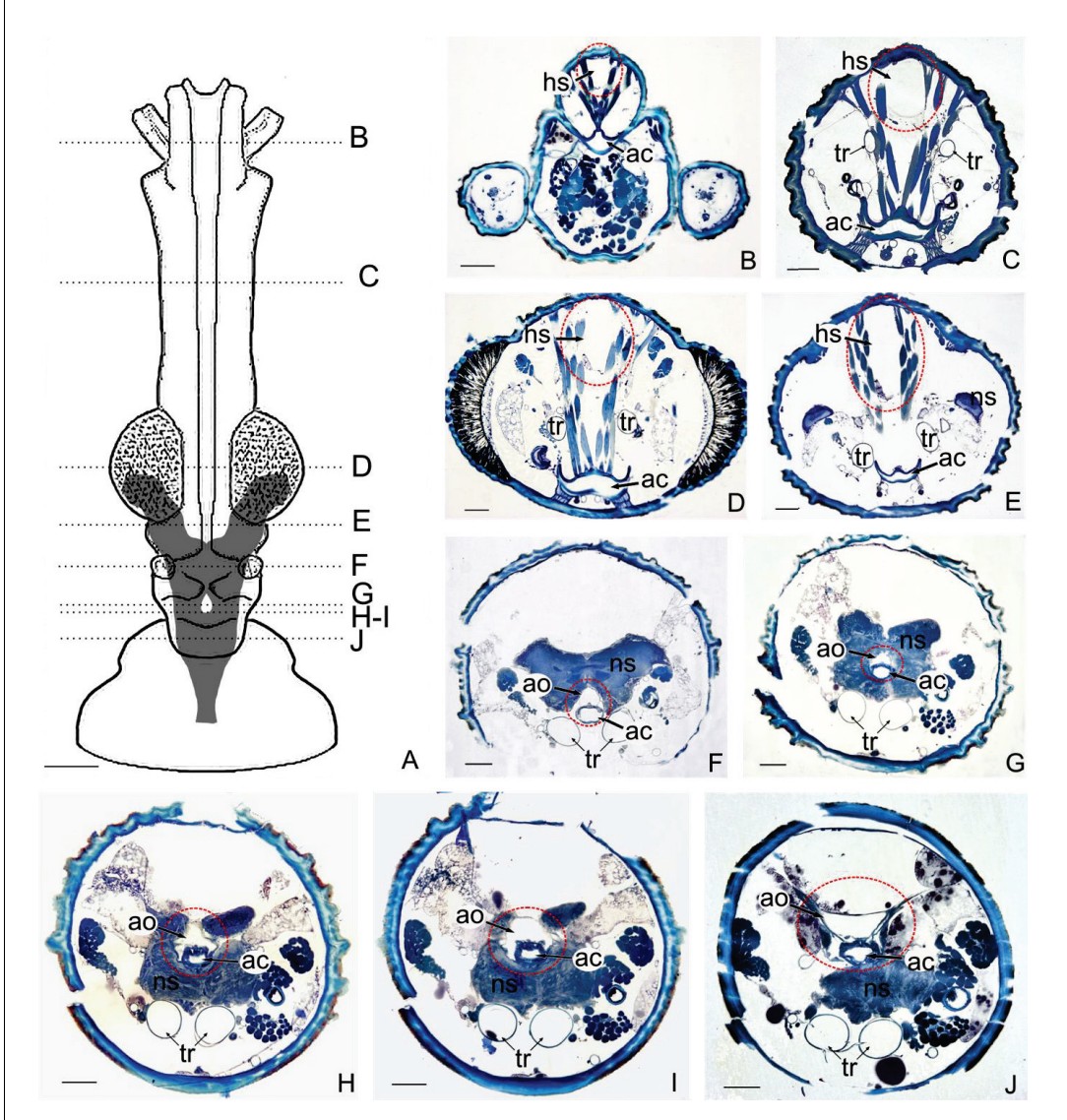

**Figure 3.** Anatomical organisation of the head of a kissing bug. (A) Diagram of the head of *Rhodnius prolixus* showing the position of the brain. (B–J): Photomicrographs of frontal sections of the head at different levels indicated in the diagram A. Dotted red lines show the position of the haemolymphatic sinus (B–E) and the aorta (F–J). Note the close association between the aorta and the alimentary canal (B–J) and the dorsal haemolymphatic sinus with the dorsal head surface (B–E) where heat transfer takes place. *ac*, alimentary canal; *ao*, aorta; *hs*, haemolymphatic sinus; *ns*, nervous system; *t*, trachea. Dotted red lines show where heat transfer takes place. Scales bars: A, 300 μm; B-J, 100 μm.
DOI: https://doi.org/10.7554/eLife.26107.011

## Alimentary canal and associated structures

In the head, the alimentary canal is associated with two muscular structures, the cibarial and pharyngeal pumps on the anterior region, and the oesophagus posteriorly (*Figures 5,6*). On the anterior region of the head, the cavity of the cibarial pump is a flattened U-shaped structure of about 150 μm large, running medially (*Figure 3A–B*). The muscles of the cibarial pump are voluminous, occupying much of the anterior part of the head (*Figure 3C–E*). They are obliquely attached to the dorsal wall of the *cibarium*. Then, the alimentary canal bends down and runs until the level of the brain (*Figure 3F*) where it bends up to go through the brain (*Figure 3H*). The oesophagus consists of a cylindrical tube of about 50 μm diameter that extends to the thorax (*Figure 3I*). The circulatory system and the alimentary canal are in close contact almost throughout the length of the head (*Figures 3*, *4C–H*). Moreover, transverse sections highlight that in some areas, the aorta surrounds

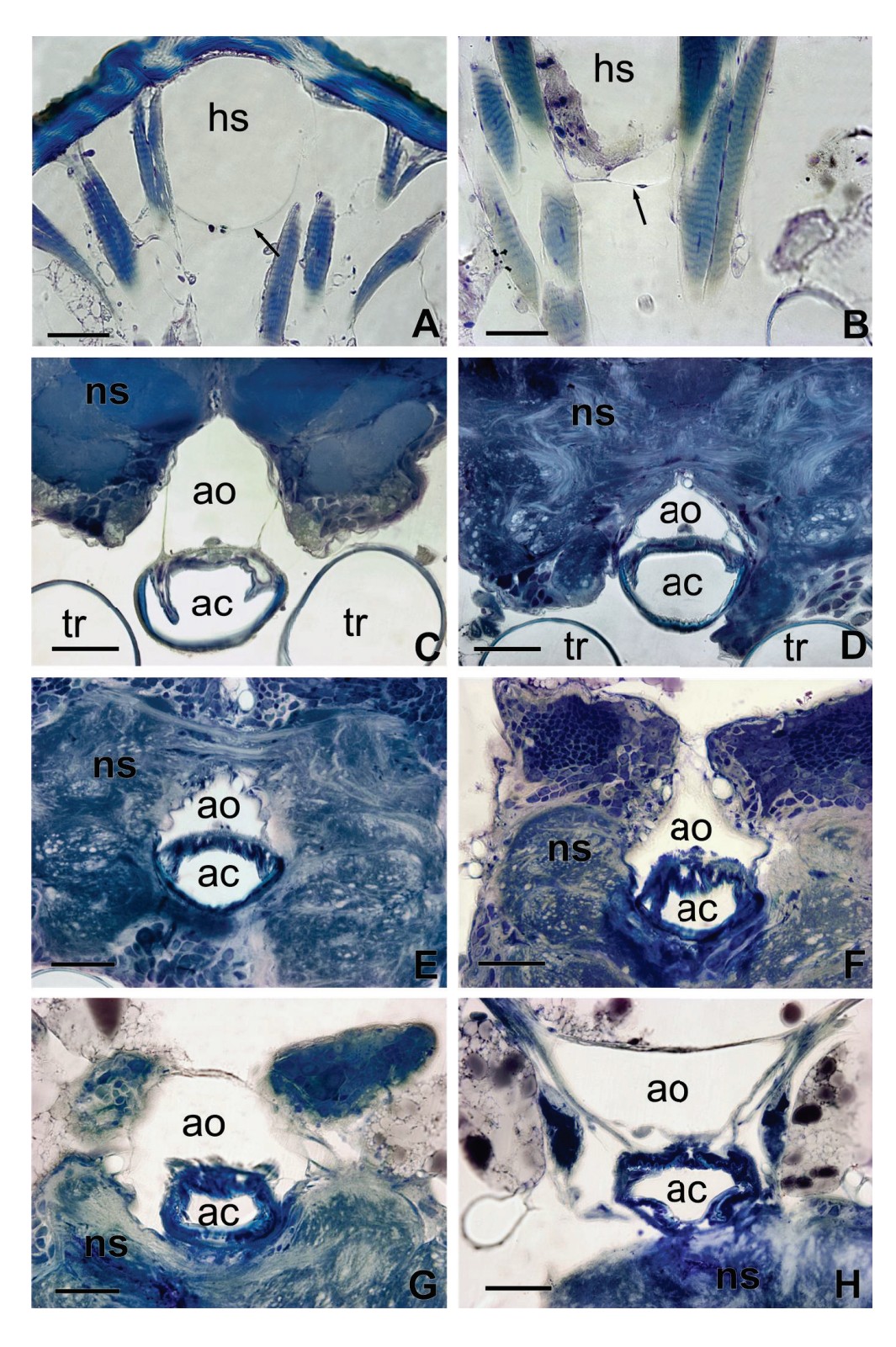

**Figure 4.** Sections of *Figure 3* showing in details the interaction between the aorta and the alimentary canal (**C–H**) and the haemolymphatic sinus with the dorsal tegument of the head (**A**). The arrow in (**A**) shows the ventral membrane delimiting the basal part of the haemolymphatic sinus. (**B**) Detail of the ventral membrane of the haemolymphatic sinus (arrow). *ac*, alimentary canal; *ao*, aorta; *hs*, haemolymphatic sinus; *ns*, nervous system; *t*, trachea. Scales bars: 100 μm.

*Figure 4 continued on next page*

*Figure 4 continued*

DOI: https://doi.org/10.7554/eLife.26107.012

lateral parts of the oesophagus (*Figure 4F–G*) making their contact even more intimate. Only thin layers of cells composing the walls of the aorta and the oesophagus separate the two cavities.

## Haemolymph circulation

As in other insects, the dorsal vessel is open at its anterior part and the circulation of haemolymph is postero-anterior. The abdominal part of the dorsal vessel is attached to the dorsal diaphragm, which delimits the visceral and the pericardial sinuses. The dorsal vessel bends down anteriorly just before entering the thorax. Haemolymph enters the dilating heart (diastole) via the ostia. In *R. prolixus*, in contrast to most insect species, all the ostia (four pairs) and alary muscles are grouped in a small section of the vessel, at the terminal segments of the abdomen (*Chiang et al., 1990*), that is where the haemolymph remains the coolest during the feeding process (*Figure 2A–B*). The haemolymph is pumped by the heart and flows throughout the aorta, until it reaches the head. At this level, the haemolymph warms up by conduction due to the close contact between the aorta and the oesophagus containing the ingested blood, which circulates in the opposite direction (anterior-posterior) (*Figures 4*, *5*).

## Muscular activity of the cibarial and pharyngeal pumps

To better understand the flow of the ingested blood inside the alimentary channel, we analysed the muscular activity of the cibarial and pharyngeal pumps using synchrotron-based X-ray imaging and computed tomography (SI *Video 2* and *Video 3*). An in vivo analysis during feeding unravelled several relevant information. First, the cibarial pump works as a piston, producing a pulsed wave of blood towards the gut. Second, the small pharyngeal pump, barely considered in the literature about kissing bugs, also shows a pulsed activity, at the same frequency, but in an opposite phase to the pharyngeal pump (*Figures 5* and *6*).

## Discussion

*Rhodnius prolixus*, as other haematophagous insects (e.g. mosquitoes, bed bugs), experiences heat stress during each feeding event, due to the engorgement of relatively big quantities of warm blood. The synthesis of heat shock proteins after a blood meal confirms heat stress during feeding, even though the increase in the HSP expression is lower than other insects (*Benoit et al., 2011*; *Paim et al., 2016*). Here, we have shown that *R. prolixus* also possesses anatomical specificities and adaptations that allow this species to minimize the heat transfer to its abdomen, thus reducing the

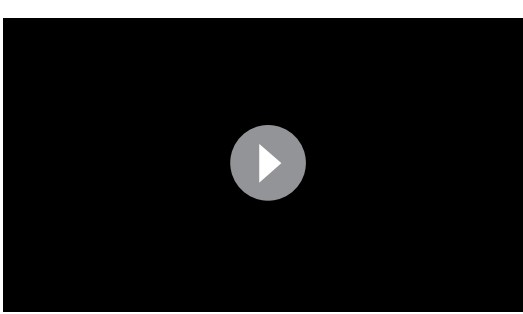

**Video 2.** X-ray synchrotron video in a sagittal view during blood intake of a fifth instar *Rhodnius prolixus* larvae showing the activity of the cibarial and pharyngeal pumps.

DOI: https://doi.org/10.7554/eLife.26107.015

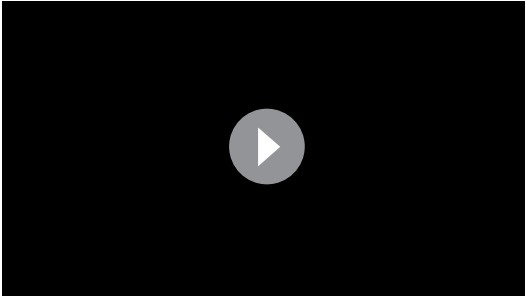

**Video 3.** X-ray synchrotron video in dorsal view during blood intake of a fifth instar *Rhodnius prolixus* larvae showing the activity of the cibarial pump as well as the tracheal network surrounding each side of the pumps and extending towards the thorax and abdomen of the insect.

DOI: https://doi.org/10.7554/eLife.26107.016

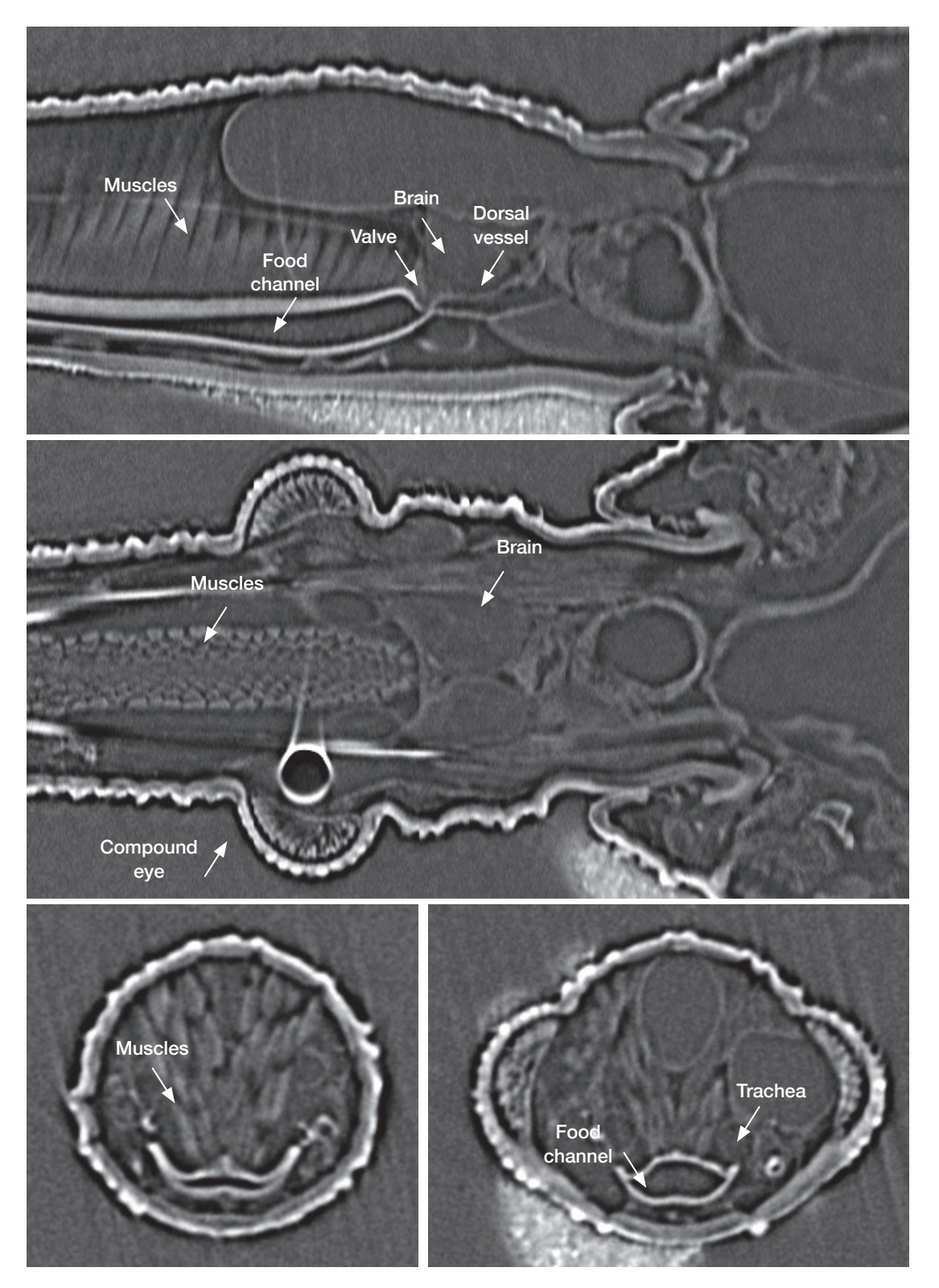

**Figure 5.** In vivo μ-CT(micro-computed-tomography scanner) imaging of a fifth instar *Rhodnius prolixus* larvae head in sagittal view (top), horizontal view (middle) and traversal view (bottom) showing the organization of the different structures in the insect head.
DOI: https://doi.org/10.7554/eLife.26107.013

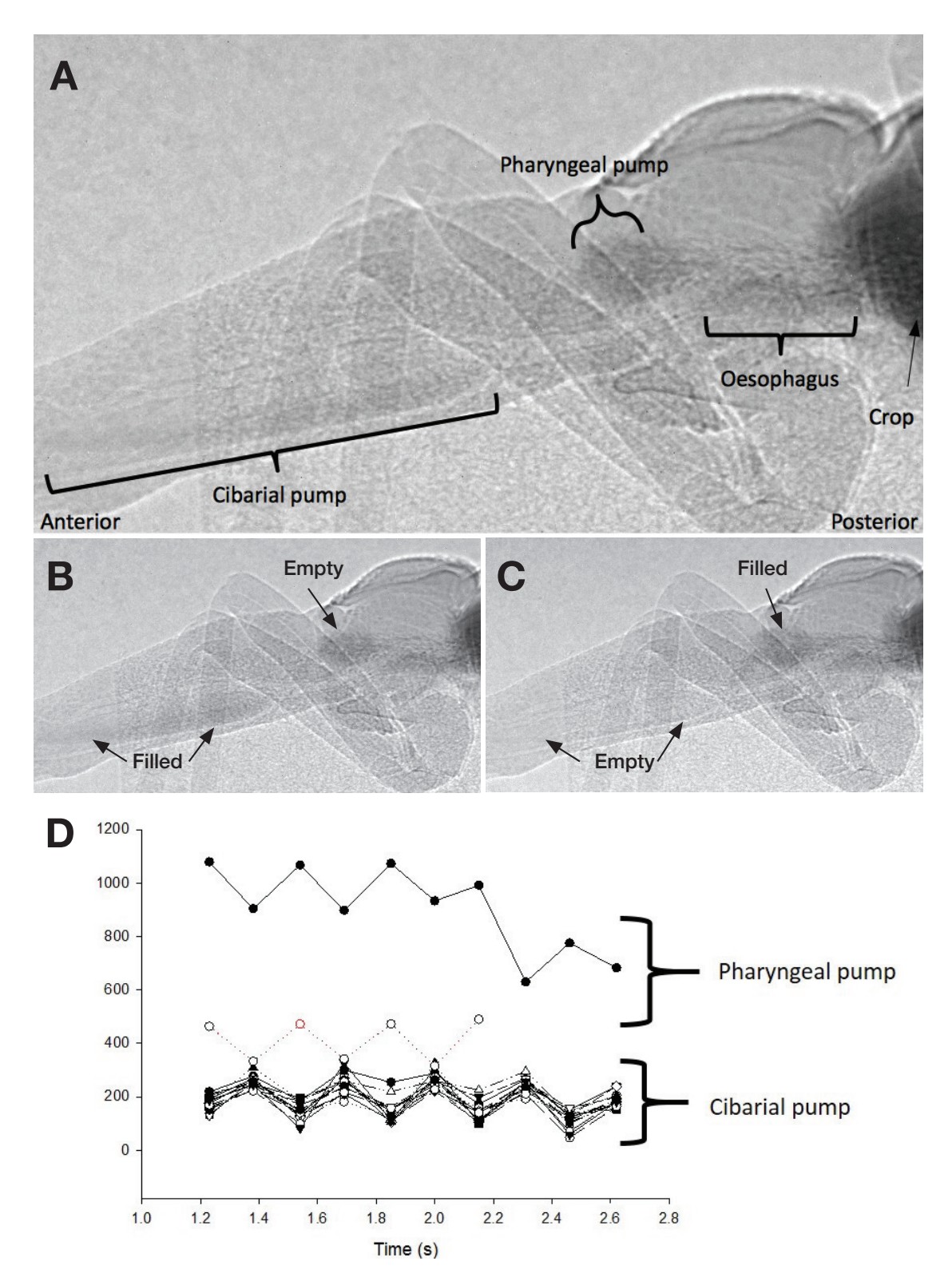

**Figure 6.** Ingestion pumps in kissing bugs. (**A**) X-ray imaging of a fifth instar *Rhodnius prolixus* larvae in sagittal view during blood-feeding. (**B**) and (**C**). Pictures showing the activity of the ingestion pumps during blood-feeding. (**D**) Activity of the cibarial and pharyngeal pumps during blood intake showing that they work in antiphase.

DOI: https://doi.org/10.7554/eLife.26107.014

heat shock associated with feeding. While the head is the most exposed to heat, the amount of heat reaching the thorax and the abdomen is dramatically reduced.

Based on our integrative study combining real-time infrared thermography with functional morphology methods, such as classical histology, X-ray in vivo-imaging and µCT, and molecular biology, we have been able to get a functional picture of the way kissing bugs quickly get rid of the excess of heat entering their body when they feed on an endothermic host. For a better perception, we built a 3-D representation of the head, thus revealing the spatial arrangement of the different structures and their relationships (*Figure 7*). Anatomically, the structures conform to a countercurrent heat exchanger between the alimentary canal and the circulatory system. The countercurrent is characterised by two currents of fluids circulating in opposite directions inside closely associated and parallel structures. The warmest fluid loses heat by conduction transferring it to the coolest one. In the area where the aorta is in close contact with the oesophagus, cool haemolymph coming from the end of the abdomen withdraws heat from the just ingested blood that is flowing into the alimentary canal in the opposite direction. Then, the haemolymph flows in the semi-open space (i.e. haemolymphatic sinus) and bathes the warm muscles of the ingestion pump. Finally, in the anterior part of the head, the haemolymph is conducted to project through the haemocoel, thus circulating in close contact with the body wall, releasing heat towards the environment (*Figure 7*). The continuous flow of both fluids is required for an efficient heat exchange. This appears to be assured by the coordinate action of the two ingestion pumps, that iscibarial and pharyngeal, which contraction on opposite phase results in a continuous flow of warm blood to the gut, resembling the Windkessel effect of the vertebrate aorta (See SI *Video 1*). On the other hand, the peristaltic contracting waves of the dorsal vessel do the same with the cool haemolymph, but in the opposite direction. The close contact between the aorta and the oesophagus facilitates the exchange of heat, which returns to the head. Consequently, only a relatively small amount of heat leaves the head (*Figure 2A*). In this way, by sequestering the heat in the head, this countercurrent heat exchanger allows the insect to minimise heat transfer to the thorax and the abdomen, thus maintaining a regional heterothermy (*Figure 2*). Our proposed model for the countercurrent exchanger here is based on both anatomical and experimental evidence.

Countercurrent heat exchangers are present in some insect species, where they modulate the flow between the thorax and the abdomen, of heat excess produced by the activity of flight muscles. In *Cuculiinae* winter moths, two heat exchangers are localised in the thorax and at the junction between the thorax and the abdomen. These exchangers enable these insects to fly at very low $T_a$ by sequestering heat in their thorax. Air sacs also help to restrain heat propagation to the abdomen (*Heinrich, 1987*). In bumblebees, the exchanger is located in the narrow passage of the petiole and helps them to exchange heat via the haemolymph between the thorax and the abdomen (*Heinrich, 1976*). As in previous examples, *Xylocopa* carpenter bees produce endogenous heat during flight. The countercurrent exchanger occurs in the petiole area where the aorta presents many loops thus facilitating heat dissipation via the abdomen (*Heinrich and Buchmann, 1986*). Robber flies (Diptera: Asilidae) also use their abdomen as a heat dissipater (*Morgan and Shelly, 1988*). All these anatomical particularities help these insect species to maintain a regional heterothermy, and has been speculated that this could protect critical organs (e.g. gonads) from an eventual thermal stress due to extreme temperature, keeping others (e.g. muscles) at optimal higher temperatures. The heat-exchange system of *R. prolixus* is unique, in the sense that it is located in the head and that it is involved in the management of the heat flow derived from the food intake.

Countercurrent heat exchangers are not restricted to invertebrates, they also occur in vertebrates such as birds (*Arad et al., 1989*), fishes (*Carey et al., 1971*; *Stevens et al., 1974*), and mammals (*Scholander and Schevill, 1955*). In these animals, heat transfer occurs between arteries reaching zones of potential heat loss (e.g. appendices) and veins carrying cold blood, in order to recuperate heat back to the body core (e.g. artic birds and mammals) or to active muscles (e.g. fishes).

The heat exchanger found in the head of *R. prolixus* is associated to other features that facilitate heat dissipation during feeding, as the elongated form of the head, which increases the surface/volume ratio and the relatively glabrous cuticle, typical of hemipterans. For instance, bumblebees show a regional difference in depth of insulation: the regions where heat needs to be stored are densely piled (thorax, dorsal part of the abdomen) whereas the ventral part of the abdomen, which serves as a thermal radiator, is completely free on hair (*Heinrich, 1976*). In some syrphid flies that produce heat endothermically with thoracic muscles, the removal of hair from the thorax leads to an increase

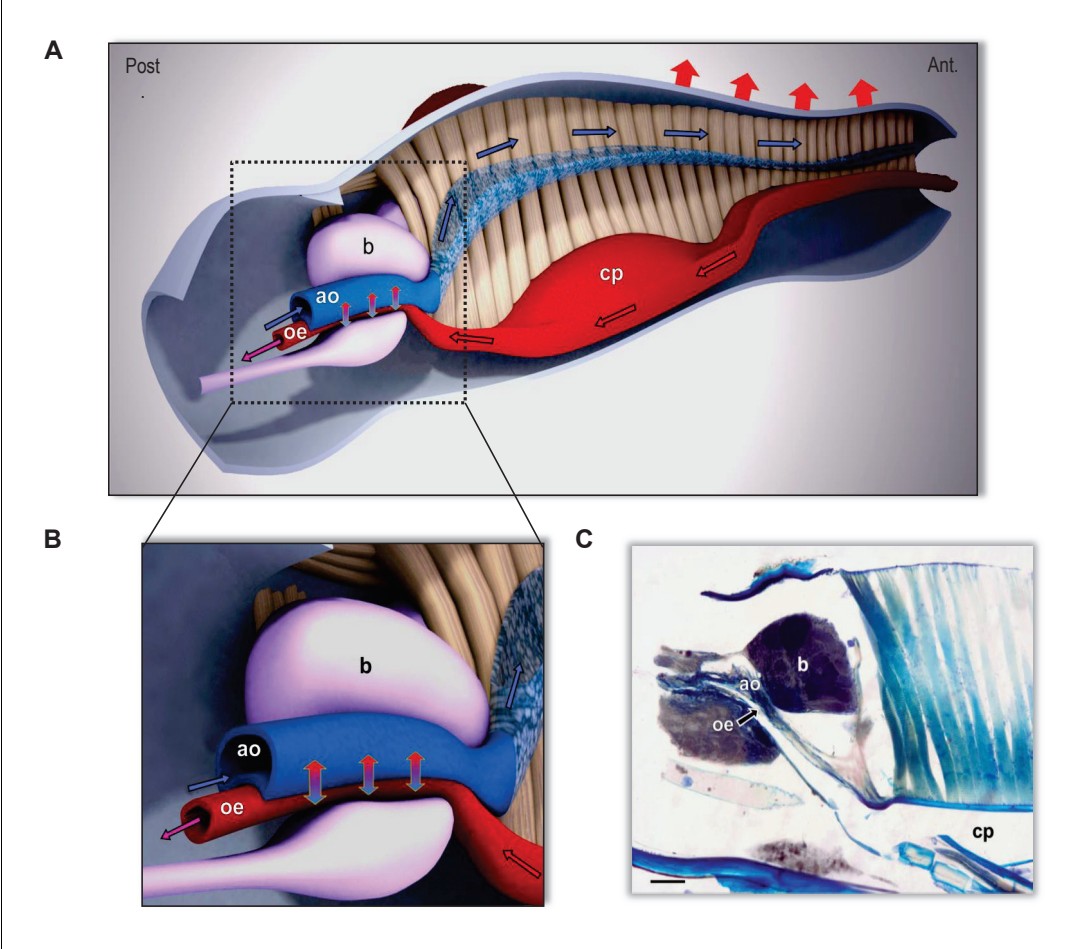

**Figure 7.** The countercurrent heat-exchanger in the head of kissing bugs. (**A**) Sagittal section of the head of *Rhodnius prolixus* (scale bar, 100 µm). (**B, C**) 3-Dimensional reconstruction of the head based on frontal and sagittal sections. A: Detail of the countercurrent heat exchanger at level of the brain. (**B**) General view of the head and circulation routes of ingested blood (red) and haemolymph (blue). Note the close association between the aorta (ao) and the alimentary canal (oe) at the level of the brain (b). In beige, the muscles of the cibarial pump (cp). Directions of flowing fluids are indicated by arrows: red/black arrows indicate the direction of the ingested blood circulating in the alimentary canal; blue/black arrows indicate the flow direction of the haemolymph. Red arrows on top indicate the heat loss to the environment by both convection and radiation. Vertical blue/red arrows indicate the countercurrent heat transfer between aorta and oesophagus fluids and the consequently refreshed blood that continues toward the thorax (purple arrow). The exact location of the system is behind the compound eyes, as can be appreciated in *Figures 2–4*.
DOI: https://doi.org/10.7554/eLife.26107.017

of the cooling rate of about 30% compared to hairy flies of the same species (*Heinrich, 1987*). Winter moths that need to keep heat to be able to fly in the cold maintain a high thoracic temperature and minimise heat loss by means of a thick insulated pile (*Heinrich, 1987*).

Moreover, the heart of *R. prolixus* is unusual in that all eight *ostia* are grouped in a short section at the very end of the abdomen (segment VII) (*Chiang et al., 1990*), where the temperature of the haemolymph is the coolest (*Figure 2A and B*) inside the insect body. In other Hemiptera species, ostia are grouped at the posterior region of the abdomen (*Hinks, 1966*). In the case of the haematophagous bugs *R. prolixus* and *Triatoma infestans*, they are located further in the terminal part abdomen, grouped in the VII and VIII abdominal segments (*Hinks, 1966*; *Barth, 1980*; *Chiang et al., 1990*). This is not the case for all blood-sucking insects. For instance, *Stomoxys calcitrans* has three pairs of ostia which spanned five abdominal segments (*Cook and Meola, 1988*) and in *Anopheles gambiae* hemolymph enters the heart through six pairs of incurrent abdominal ostia and one pair of ostia located at the thoracic-abdominal junction (*Glenn et al., 2010*).

It is worth mentioning also that, as mosquitoes, bugs eliminate drops of urine during feeding (SI *Video 1*). Although the size and the time that these drops remain attached to the tip of the

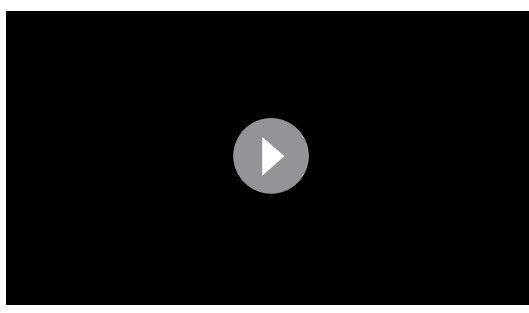

**Video 4.** Three-dimensional reconstruction of the tracheal system (in pink) in *Rhodnius prolixus'* head showing the two big trachea on each side of the cibarial pump (in blue) as well as secondary and tertiary trachea networks surrounding the pump. Note that the tracheal system extends beyond the prothorax (see *Ramírez-Pérez, 1969* for a complete overview of the tracheal network all along the insect's body).
DOI: https://doi.org/10.7554/eLife.26107.018

abdomen are both reduced, and evaporative cooling is hardly probable to occur as in *Anopheles* (*Lahondère and Lazzari, 2012*), these drops may reduce the temperature of the terminal part of the abdomen, precisely in the area where the haemolymph enters the heart to be pulsed forwards.

The aorta is open on its dorsal part in the cephalic portion and goes through the cibarial pump muscles. The warmed haemolymph is thus poured around these muscles that are also warmed by both contractions and direct contact with the alimentary canal which contains the circulating warm blood. The wide sinus, in which the insect haemolymph circulates in contact with the insect integument, probably facilitates heat loss by convection and radiation. Finally, two tracheas extend along the head, next to both the pharyngeal pump (*Figure 3C–D*) and the alimentary canal (*Figure 4C–D*, SI *Video 3* and SI *Video 4*) (*Ramírez-Pérez, 1969*). These tracheas may also help decrease the temperature of the head by circulating air and by internal evaporative cooling. Internal evaporative cooling is a common strategy in species, such as plant juice feeders (*Prange, 1996*), which can obtain water easily. In the tsetse fly *Glossina morsitans*, a decrease of 2°C of the $T_b$ occurs when the $T_a$ exceeds the thermal upper limit of tolerance thanks to evaporation via the spiracles (*Edney and Barrass, 1962*). The process of evaporative cooling via the ventilatory system is also documented in grasshoppers and beetles with a decrease of the $T_b$ up to 8°C (*Prange, 1990*). In *R. prolixus* during feeding, there is an increase in the rate of exchange of $CO_2$, $O_2$, and water vapour: this is thought to be linked with spiracle opening (*Leis et al., 2016*). If the hypothesis of heat loss via the trachea is true in *R. prolixus*, it could resemble to respiratory countercurrent mechanisms found in the nose of mammals, birds, and reptiles. In these animals, the heat exchanger cools the blood flowing to the brain to reduce overheating (*Tattersall et al., 2006*). This hypothesis is of course quite speculative, but deserves to be further explored.

The present work has shed some light on different mechanisms, adaptations and strategies helping *R. prolixus* to avoid the thermal stress associated with blood-feeding on endothermic animals. The 3-D reconstruction of the head based on histological sections and imaging techniques support the hypothesis for a countercurrent occurring in the head of this species helping the insect to regulate its own temperature during blood-feeding.

Until now, the only thermoregulatory mechanism associated with blood-feeding known in insects, was evaporative cooling
performed by *Anopheles* mosquitoes (*Lahondère and Lazzari, 2012*). Our work in *R. prolixus* not only unravelled a new species possessing thermoregulatory abilities, suggesting that it may be also present in other haematophagous insects, but also revealed a new original mechanism, based on morphological and physiological specific adaptations. It is possible that countercurrent heat exchangers could be more frequent and diversified than commonly thought.

## Materials and methods

### Insects

Experimental bugs came from laboratory colonies of *Rhodnius prolixus* Stål (1859) (Heteroptera: Reduviidae: Triatominae), reared at the Insect Biology Research Institute (IRBI, Tours, France), the Universidade Federal de Minas Gerais (Brazil) and at the Department of Physiology, University of Saskatchewan (Canada). Insects were maintained under a 12:12 hr light/dark regime at 25 ± 1°C and 60–70% relative humidity. Insects are fed weekly on heparinised sheep blood via an artificial feeder (*Núñez and Lazzari, 1990*). Fifth-instar larvae were isolated after ecdysis and kept starved for 8–12

days until being used for thermographic experiments, X-ray imaging and computed tomography. Unfed adults were used for histological preparations because of their bigger size.

## Thermographic recordings and data analysis

Infrared thermography is a non-invasive technique that accurately measures the temperature at the surface of the insect body during the entire process of blood-feeding.

### Thermographic camera

Temperature recordings were made with a thermographic camera (PYROVIEW 380L compact, DIAS infrared GmbH, Germany; spectral band: 8–14 mm, uncooled detector 2D, 384 × 288 pixels) equipped with a macro lens (pixel size 80 µm; A = 60 mm; 30° x 23°). Thermal data were acquired and recorded with the RTV Analyser RT software (IMPAC Systems GmbH, Germany). The emissivity was fixed at 0.98, as determined by *Stabentheiner and Schmaranzer, 1987*) for insect cuticle.

### Thermographic recordings

We first evaluated the impact of the blood meal temperature ($T_{blood}$) on body temperature ($T_b$) of fifth-instar larvae taking a blood meal at either 32°C, 37°C or 42°C (±1°C) in a room at 22 ± 1°C (*Figure 1A*). Then, based on these results, we performed a set of feeding experiments using different combinations of ambient temperature ($T_a$) and blood temperature ($T_{blood}$): [($T_a$ = 16°C) * ($T_{blood}$ = 32°C or 37°C)]; [($T_a$ = 21°C) * ($T_{blood}$ = 32°C, 37°C or 42°C)]; ($T_a$ = 26°C) * ($T_{blood}$ = 37°C or 42°C)]; ($T_a$ = 31°C) * ($T_{blood}$ = 37°C or 42°C)] (±1°C for all the temperatures) (*Figure 1B*). A total of 27 bugs were used to test the nine temperature combinations (three per combination; see below for data analysis). As control, thermographic recordings of resting insects were performed at 25°C.

Both ambient temperature and blood temperature were measured with a thermocouple-thermometer (Hanna, HI9043, accuracy 0.1°C, sensor dia. <1 mm). Measurements of larvae body temperature were regularly made by quickly inserting the thermocouple into the abdominal cavity of the insect.

Finally, to understand the implication of haemolymph circulation in the insect's body warming during blood-feeding, the dorsal vessel of five insects were cut at the level of the fifth abdominal tergite. A similar number of bugs were sham operated, that is the body wall was cut, but the dorsal vessel remained intact. Briefly, a small window was opened to expose the dorsal vessel. A small section of the vessel was removed, by doing two cuts separated by about 1 mm. Then, the cuticle was replaced, in order to close the window and bugs let recovering for 48 hr, before feeding them for thermographic measurements.

### Thermal data analysis and statistics

Thermographic images were captured during the whole feeding process (*Figure 1—figure supplement 1*), excluding the probing times when the insect started the blood meal intake, pumping irregularly, and after the feeding process was almost finished (i.e. about 3 min before the end of the blood-meal, depending on the blood temperature). For the first experiments, the temperatures of the centre of the proboscis ($T_p$), head ($T_h$), thorax ($T_{th}$), and abdomen ($T_{ab}$) of each individual were measured frame by frame from the recordings (RTV analyser software). We also measured the temperature gradient all along the insect body to evaluate patterns of heat diffusion.

For the second series of experiments (different combinations of ambient and blood temperatures), we randomly selected ten frames for each insect and measured the temperature of each body part as in the previous experiment. In this case, we compared the body temperatures of the experimental groups with a *two-way* ANOVA (factors: body part temperatures and $T_{blood}$) whenever the data were normally distributed and had homogeneous variances. Tukey tests were conducted *post hoc* to identify significant pair-wise comparisons. For the experiment in which the dorsal vessel was severed, temperatures of different body parts were measured during the feeding. Temperatures were then averaged within body parts for each of the five insects and the standard deviations were calculated. All graphs and statistics were done using SigmaPlot V.12 (Systat Software Inc, San José, California, USA) and R (Development Core Team, http://www.R-project.org).

## Histology and microscopy

Light microscopy was performed on insect heads following the procedure described by Reisenman and collaborators (*Reisenman et al., 2002*). Briefly, freshly decapitated heads were fixed for 3 hr in a mixture of 2.5% glutaraldehyde and 2.0% paraformaldehyde in phosphate buffer (pH 7.3) with glucose and $CaCl_2$ added. After gradual dehydrated in 100% ethanol, they were embedded via propylene oxide in Durcupan ACM (Electron Microscopy Sciences no. 14040). We used glass knives mounted on a motorised microtome to serially section blocks at 2–5 μm. Sections were stained on a hot plate with Toluidine Blue-Basic Fuchsin and mounted on a slide with DPX (Electron Microscopy Sciences no. 13510). Photomicrographs were adjusted for brightness and contrast by using Adobe Photoshop CS2. A total of 10 individuals were prepared, sectioned, stained and analysed.

## Synchrotron-based X-ray imaging and computed tomography

### Synchrotron-based fast X-ray imaging

Synchrotron x-ray imaging was used to observe the internal structure of *R. prolixus*. Experiments were performed on the BioMedical Imaging and Therapy - bending magnet (BMIT-BM) beamline 05B1-1 at Canadian Light Source (CLS) (*Wysokinski et al., 2007*). The experiments were done 25.5 m from the source.

Starved fifth-instar *R. prolixus* were presented with warm, defibrinated rabbit blood (Cederlane, Ontario, Canada) labelled with organic iodide (Optiray 240, Tyco Healthcare, New Jersey) making the blood opaque to X-rays. Blood was placed in an artificial feeder and covered with a thin latex membrane that allowed the insects to pierce and feed. Insects were placed in a plastic holder that allowed them to move freely. One end of the holder was open and in direct contact with the latex membrane of the artificial feeder. The artificial feeder and the animal holder were placed on a motorised stage so we could remotely control the insect's position in the X-ray beam. The images were captured using a high-resolution X-ray beam monitor (AA-40 Hamamatsu Photonics) optically coupled to a charge-coupled device (CCD) detector (C9300-124 Hamamatsu Photonics). The effective pixel size of the beam monitor- CCD camera combination is $4.3 \times 4.3$ μm. The X-rays were converted to visible light by a 10 μm-thick powder scintillator P43 (Gd2O2S:Tb). The monochromatic X-ray flux at BMIT-BM line is not high enough to collect data with a speed of at least 6–10 fps (frames per second). For that reason, we opted for using filtered white beam. The preliminary experiments showed that insects were feeling the presence of the X-ray beam when the dose rate was too high. They stopped feeding within 10 to 30 s after the beginning of the X-ray exposure and were moving to the sides of the holder trying to get out of the X-ray beam. The animals exposed to the highest dose rates that the X-ray detector could generate, died after a few minutes of continuous exposure to the beam. In order to avoid the effects described above we changed the average energy and the flux of the X-ray arriving at the sample location by proper selection of the metal filters inserted in the beam (0.055 mm Cu and 5 mm Al). As a result of that, the dose rate became much smaller and the animals did not react to the presence of the X-ray beam. The filtration of the beam limited the speed at which the data could be collected to about 6.5 fps. For the period of time when insects initiated feeding, the X-rays were blocked by fast (20 ms actuation time) mechanical shutter. After an insect had fed for 1 to 3 min (i.e. when it has finished probing and began feeding), the shutter was opened and the animal was imaged (*Sena et al., 2014*). A total of 16 individuals were used, some of them were analysed more than once.

### Synchrotron-based X-ray phase contrast computed tomography

X-ray phase contrast computed tomography (CT) scan was performed using monochromatic 14.0 keV (λ = 0.089 nm) X-rays, selected using the beamline Bragg double-crystal monochromator (Si2,2,0). For the CT scan the distance between sample and detector was 100 cm in order to boost the phase contrast in the images and to gain some edge enhancements in the collected images. A total of 1800 images were collected over 180 degrees' rotation, keeping the angular increment constant (0.1 deg). At each angular position two images with exposure time in the range of 200–400 ms were recorded and added together to improve the signal to noise ratio in the collected data. Transversal and sagittal sections were done with ImageJ (NIH, USA).

Fourteen starved fifth instar *R. prolixus* were injected with 8 to 10 μl of fixative solution though the joint connecting the abdomen to the coxa of the third or secondleg. Fixative stock solution was

made by mixing 80 ml ethanol 100%, 8 ml acetic acid glacial and 2 ml of 4% formaldehyde. Insects were incubated at room temperature for 24 hr, immersed in 2% agar in PBS, and placed in a plastic cylindrical holder filled with 2% agar. Next the insect (in the holder) were placed on a motorised stage and imaged.

## Data analysis

Since improvement of the absorption x-ray contrast was achieved by labelling the rabbit blood with organic iodide (iodine has much higher x-ray absorption than the animal tissue), the intensity of the X-ray reaching the detector was inversely related to the amount of rabbit blood present in the alimentary canal. Thus, the alimentary canal full of rabbit blood appears relatively dark. These changes in contrast of the images were used to study the anatomy of the alimentary canal and the function of the pharyngeal and cibarial pumps driving the blood meal.

## Three-dimensional reconstruction of the head

The 3-D reconstruction of the head structure and morphology was performed by compiling both transversal and sagittal serial histological sections and using 3ds Max software (Autodesk).

## RNA extraction, cDNA synthesis and qPCR

Insects that had their dorsal vessel severed or sham operated (body wall opened and closed again) were fed on blood using an artificial feeder (blood temperature 39°C, environment 28°C). Even though sham bugs fed a bit faster than bugs with their dorsal vessel severed, all bugs reached about five times their initial weight. After 2 hr, insects were dissected to collect their midguts for individual RNA extraction using the Nucleospin RNA II Kit (Macherey-Nagel). RNA was treated with DNase according to the manufacturer's instructions and eluted in 20 μL of ultra-pure RNase-free water. Then, purified RNA was quantified by measuring 260 nm wavelength absorbance and 0.5 μg was used for cDNA synthesis with 0.5 μg of random hexamers (Promega) using the M-MLV reverse transcriptase system (Promega) in a final volume of 25 μL. The cDNA was used in qPCR assays, using the StepOne Plus real time quantitative PCR system (Applied Biosystems) to evaluate HSP70 and HSP90 expression. Each reaction was run in duplicate and contained 20 ng of cDNA, specific primers for HSP70 (forward: 5'-gaaatcgtactggttggtgga-3', reverse: 5'-cgccataggctacagcttca-3') or HSP90 (forward: 5'-ggacccatcaagactggaga−3', reverse: 5'- agcaatggttcccagattgt - 3') at a final concentration of 300 nM each and 5 μL of 2x Power SYBR Green PCR Master Mix (Applied Biosystems) in a final volume of 10 μL. Amplification conditions were 95°C for 10 min, 40 cycles of 95°C for 15 s and 60°C for 1 min. A reverse transcription negative control (without reverse transcriptase) and a non-template negative control were included to confirm the absence of genomic DNA and to check for primer-dimer or contamination in the reactions, respectively. To ensure that only a single product was amplified, the melting curve was analysed. The relative amount of gene product in each sample was determined with 2-ΔΔCt method (*Livak and Schmittgen, 2001*) using α-tubulin (forward: 5'-tttcctcgatcactgcttcc-3', reverse: 5'-cggaaataactggggcataa- 3') as a reference gene (*Paim et al., 2012*).

After verification of normality and homoscedasticity by means of the Shapiro-Wilk and F-test (GraphPrism), HSP expression levels (n = 5 individuals per group) were compared by means of a Student *t*-test for independent samples and the significance of the difference evaluated as two-tailed p-value.

## Acknowledgements

We are indebted to Carole Labrousse for taking care of the *Rhodnius prolixus* colony at the University of Tours, France. We are grateful to Eugenio Insausti and Orhan Yilmaz for their help on the 3-D reconstructions and to Clément Vinauger for his valuable comments on the manuscript, helpful discussions and help with the movies formatting. We would like to thank Raymond B Huey for his helpful comments and edits on the manuscript. We also thank Catherine Bourgouin and Michael Lehane for their helpful insights on an earlier version of the manuscript. Joshua B Benoit and David Denlingen, together with anonymous reviewers and editors made valuable suggestions that helped us to improve the manuscript. This work received financial support from the Agence Nationale de la

Recherche (grant ANR-08-MIE-007, EcoEpi), the Centre National de la Recherche Scientifique, and the University of Tours (France), the Conselho Nacional de Desenvolvimento Científico e Tecnológico, the Coordenação de Aperfeiçoamento de Pessoal de Nível Superior, and the Fundação de Amparo à Pesquisa do Estado de Minas Gerais (Brazil), a Discovery operating grant from the Natural Sciences and Engineering Research Council of Canada awarded to JPI and part of the research described in this paper was performed at the BMIT facility at the Canadian Light Source, which is supported by the Canada Foundation for Innovation, Natural Sciences and Engineering Research Council of Canada, the University of Saskatchewan, the Government of Saskatchewan, Western Economic Diversification Canada, the National Research Council Canada, and the Canadian Institutes of Health Research.

## Additional information

### Funding

| Funder | Grant reference number | Author |
| --- | --- | --- |
| Université François-Rabelais | | Chloé Lahondère<br>Claudio R Lazzari |
| Conselho Nacional de Desenvolvimento Científico e Tecnológico | | Rafaela MM Paim<br>Marcos H Pereira |
| Coordenação de Aperfeiçoamento de Pessoal de Nível Superior | | Rafaela MM Paim<br>Marcos H Pereira |
| Fundação de Amparo à Pesquisa do Estado de Minas Gerais | | Rafaela MM Paim<br>Marcos H Pereira |
| Natural Sciences and Engineering Research Council of Canada | Discovery operating grant | Juan P Ianowski |
| Agence Nationale de la Recherche | ANR-08-MIE-007 EcoEpi | Claudio R Lazzari |
| Centre National de la Recherche Scientifique | | Claudio R Lazzari |

The funders had no role in study design, data collection and interpretation, or the decision to submit the work for publication.

### Author contributions

Chloé Lahondère, Conceptualization, Data curation, Software, Formal analysis, Investigation, Visualization, Methodology, Writing—original draft; Teresita C Insausti, Formal design, Formal analysis, Supervision, Investigation, Visualization, Methodology, Writing—review and editing; Rafaela MM Paim, Marcos H Pereira, Writing—review; Xiaojie Luan, Software, Investigation, Visualization, Formal design, Formal analysis, Writing-original draft; George Belev, Software, Investigation, Visualization, Methodology, Formal analysis, Writing-original draft; Marcos H Pereira, Resources, Supervision, Validation, Investigation,; Juan P Ianowski, Resources, Software, Formal design, Formal analysis, Supervision, Validation, Investigation, Visualization, Methodology, Writing—review and editing; Claudio R Lazzari, Conceptualization, Resources, Formal design, Formal analysis, Supervision, Funding acquisition, Validation, Investigation, Visualization, Methodology, Writing—original draft, Project administration, Writing—review and editing

### Author ORCIDs

Claudio R Lazzari (iD) https://orcid.org/0000-0003-3703-0302

### Decision letter and Author response

Decision letter https://doi.org/10.7554/eLife.26107.021
Author response https://doi.org/10.7554/eLife.26107.022

## Additional files

### Supplementary files

• Supplementary file 1. *Two-way* ANOVA tables for the analysis of the impact of the temperature of the blood ($T_{blood}$) and the environmental temperature on the different body parts of *Rhodnius prolixus* during feeding. (See also *Figure 2—figure supplement 1*). From top to bottom, $T_a$ = 16°C, 21°C, 26°C or 31°C.

DOI: https://doi.org/10.7554/eLife.26107.019

• Transparent reporting form

DOI: https://doi.org/10.7554/eLife.26107.020

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
