## [Decision Letter]

Thank you for submitting your article "Countercurrent heat exchange and thermoregulation during blood-feeding in kissing bugs" for consideration by *eLife*. Your article has been favorably evaluated by Diethard Tautz (Senior Editor) and four reviewers, one of whom, Joshua B Benoit (Reviewer #1), served as Guest Reviewing Editor. The following individual involved in review of your submission has agreed to reveal his identity: David Denlinger (Reviewer #4).

The reviewers have discussed the reviews with one another and the Reviewing Editor has drafted this decision to help you prepare a revised submission.

General assessment:

The reviewers consider this is an interesting paper with elegant experiments that presents convincing results of structural evidence for a counter-current heat exchange, which creates a heat gradient from the head to abdomen. Addition of the dorsal vessel severing provides further evidence for this cooling system. From these studies, the authors conclude that this allows for the bloodmeal to be rapidly cooled and protects from heat stress. The general consensus from the reviewers is that this is likely important and may be critical to the biology of kissing bugs. However, the concern is that results do not provide evidence that this system would be protective. Specifically, kissing bugs usually feed while standing on the warm host, which means the data for kissing bugs held at 31C is most ecologically relevant. Under this treatment, the exchange system would only account for a 2-4C change in temperature, which may not cause physiological or molecular changes. The inclusion of direct evidence, such as gene expression changes following impairment of this system, would solidify the conclusions of this study. Overall, the study is important by demonstrating an interesting cooling system that may protect kissing bugs from blood-induced heat stress.

Essential revisions:

Although the set-up and experimental work is highly appreciated, the presented data, in the view of the reviewers, does not yet fully support the conclusion that this system will protect the kissing bugs from feeding induced heat.

1) The study would be strongly improved by experimental evidence that this heat exchange system does protect kissing bugs from heat stress at either the molecular and physiological levels. This issue could be alleviated by the following:

a) Measure heat shock protein expression in the gut of kissing bugs with and without dorsal vessel severing to verify increased heat stress due to a lack of the counter-current heat exchange.

b) This should be done with ecologically relevant levels (31C experiments) that mostly likely mimic the natural feeding state of this insect.

2) The reviewers are concerned with the small sample size, suggesting the statistics are not robust and non-parametric methods should be used.

3) The entire manuscript needs to be edited to remove "casual" language.

[Editors' note: further revisions were requested prior to acceptance, as described below.]

Thank you for resubmitting your work entitled "Countercurrent heat exchange and thermoregulation during blood-feeding in kissing bugs" for further consideration at *eLife*. Your article has been favorably evaluated by Diethard Tautz (Senior Editor) and three reviewers, one of whom served as Guest Reviewing Editor.

The manuscript has been improved but there are some remaining issues that need to be addressed before acceptance, as outlined below:

The HSP70 does indeed indicate that there is likely a protective effect, which is a major improvement. The authors mentioned that the cooling would also prevent cleptohaematophagy. The inclusion of this data would make this paper much more interesting.

---

## [Author Response]

Essential revisions:Although the set-up and experimental work is highly appreciated, the presented data, in the view of the reviewers, does not yet fully support the conclusion that this system will protect the kissing bugs from feeding induced heat.

We appreciate the comment. This new version includes novel experiments testing the protective role against thermal stress (HSP expression). We verified a significant increase in HSP70 in bugs with severed dorsal vessels, as compared to sham individuals. Just for information, since not included in this manuscript, we have also found that bugs which body temperature is some degrees above the environment are bitten by starved conspecific, which perform cleptohaematophagy and cleptohaemolymphophagy. So, the protective role of the cooling system seems go beyond protection against thermal stress in this gregarious species.

1) The study would be strongly improved by experimental evidence that this heat exchange system does protect kissing bugs from heat stress at either the molecular and physiological levels. This issue could be alleviated by the following:a) Measure heat shock protein expression in the gut of kissing bugs with and without dorsal vessel severing to verify increased heat stress due to a lack of the counter-current heat exchange.b) This should be done with ecologically relevant levels (31C experiments) that mostly likely mimic the natural feeding state of this insect.

Following the advice of reviewers, we measured the expression of HSP70 and HSP90 in bugs which dorsal vessel was severed and sham individuals. The results integrated in Figure 2 and in the subsection “2. HSP expression and thermal stress”. For this, we requested the assistance of two colleagues with recognised experience on *Rhodnius* HSPs, Marcos Pereira (Prof.) and Dr. Rafaela Paim (post-doc), with whom we collaborate regularly and already published together on a related subject. They kindly accepted our invitation, performed the experiments under the requested conditions, evaluated the expression of HSP70 and HSP90 on the two groups and analysed the data. For this reason, the rest of the authors consider that their contribution is substantial enough for including them as co-authors. Marcos and Rafaela accepted co-signing the paper and they are, of course, aware of the content of the manuscript.

2) The reviewers are concerned with the small sample size, suggesting the statistics are not robust and non-parametric methods should be used.

The small sample size is due to the complexity in performing the experiments in exactly the same way. Nevertheless, in all cases, before applying parametric statistics, normality, skewness and homoscedasticity were verified to avoid any bias.

3) The entire manuscript needs to be edited to remove "casual" language.

Thank you very much for the comment. We have revised the English, in order to render the text less casual.

[Editors' note: further revisions were requested prior to acceptance, as described below.]

The manuscript has been improved but there are some remaining issues that need to be addressed before acceptance, as outlined below:The HSP70 does indeed indicate that there is likely a protective effect, which is a major improvement. The authors mentioned that the cooling would also prevent cleptohaematophagy. The inclusion of this data would make this paper much more interesting.

Concerning the relationship between thermoregulation and cannibalism in *Rhodnius prolixus* that we mentioned in our previous response to reviewers, we much appreciate the advice. Yet, we prefer keeping it apart for different reasons. First, this parallel study contains a body of information in terms of methodology, results and conclusions that will require rewriting the whole manuscript in order to not disperse the attention of the reader; second, it would mean a whole new submission that will probably require a new round of evaluation by additional reviewer, delaying publication (an issue for some of the authors) and last, but not least, additional authors should be added, with the consequent discussion about authorship order.